

# Illegal use of natural resources in federal protected areas of the Brazilian Amazon

Érico E. Kauano[1,2], Jose M.C. Silva[1,3] and Fernanda Michalski[1,4,5]

[1] Programa de Pós-Graduação em Biodiversidade Tropical, Universidade Federal do Amapá, Macapá, Amapá, Brazil
[2] Parque Nacional Montanhas do Tumucumaque, Instituto Chico Mendes de Conservação da Biodiversidade, Macapá, Amapá, Brazil
[3] Department of Geography - Geography and Regional Studies, University of Miami, Coral Gables, FL, United States of America
[4] Laboratório de Ecologia e Conservação de Vertebrados, Universidade Federal do Amapá, Macapá, Amapá, Brazil
[5] Instituto Pro-Carnívoros, Atibaia, São Paulo, Brazil

Corresponding author
Érico E. Kauano,
ericokauano@gmail.com,
erico.kauano@icmbio.gov.br

## ABSTRACT

**Background**. The Brazilian Amazon is the world's largest rainforest regions and plays a key role in biodiversity conservation as well as climate adaptation and mitigation. The government has created a network of protected areas (PAs) to ensure long-term conservation of the region. However, despite the importance of and positive advances in the establishment of PAs, natural resource depletion in the Brazilian Amazon is pervasive.

**Methods**. We evaluated a total of 4,243 official law enforcement records generated between 2010 and 2015 to understand the geographical distribution of the illegal use of resources in federal PAs in the Brazilian Amazon. We classified illegal activities into ten categories and used generalized additive models (GAMs) to evaluate the relationship between illegal use of natural resources inside PAs with management type, age of PAs, population density, and accessibility.

**Results**. We found 27 types of illegal use of natural resources that were grouped into 10 categories of illegal activities. Most infractions were related to suppression and degradation of vegetation (37.40%), followed by illegal fishing (27.30%) and hunting activities (18.20%). The explanatory power of the GAMs was low for all categories of illegal activity, with a maximum explained variation of 41.2% for illegal activities as a whole, and a minimum of 14.6% for hunting activities.

**Discussion**. These findings demonstrate that even though PAs are fundamental for nature conservation in the Brazilian Amazon, the pressures and threats posed by human activities include a broad range of illegal uses of natural resources. Population density up to 50 km from a PA is a key variable, influencing illegal activities. These threats endanger long-term conservation and many efforts are still needed to maintain PAs that are large enough and sufficiently intact to maintain ecosystem functions and protect biodiversity.

## INTRODUCTION

The Brazilian Amazon is one of the world's largest rainforest regions and plays a key role in biodiversity conservation, maintenance of ecosystem services, and storage of terrestrial carbon stocks (*Laurance et al., 2001*). In recent years, many advances have been made in combating the widespread and illegal use of the region's natural resources. Political actions based on the establishment of new protected areas (PAs), increases in law enforcement, and support for forest-based economic activities have resulted in a significant deforestation reduction in the region (*Fearnside, 2005*; *Nepstad et al., 2009*; *Silva, Rylands & Da Fonseca, 2005*). In 2010, an extensive network of PAs protected about 54% of the remaining forests of the Brazilian Amazon and contained around 56% of its forest carbon (*Soares-Filho et al., 2010*).

The creation and maintenance of PAs is the most effective way to protect vast areas of tropical forests in the Brazilian Amazon (*Dalla-Nora et al., 2014*; *Soares-Filho et al., 2010*; *Soares-Filho et al., 2006*). Recent studies have indicated that PAs can reduce deforestation and pave the way to a more sustainable use of the region's natural resources (*Barber et al., 2012*; *Nepstad et al., 2014*; *Nepstad et al., 2006*; *Nepstad et al., 2009*; *Pfaff et al., 2015*). However, despite all these recent efforts, natural resource degradation in the Brazilian Amazon is still pervasive and thus PAs are subjected to several pressures and threats. Four major factors determine the intensity of pressures on a PA: (a) accessibility; (b) local human population density; (c) management category; and (d) age of the PA.

Accessibility of PAs can be measured by evaluation of navigable rivers and roads that cross or form the boundaries of a given reserve (*Peres & Terborgh, 1995*). *Peres & Lake (2003)* estimate that much of the Amazon basin in Brazil can be accessed on foot from the nearest river or functional road and found that the density of preferred hunted species tended to decrease in areas closer to access points (e.g., roads, rivers). In Amazonia, until 1997, about 90% of deforestation was concentrated in areas within 100 km of main roads established by federal government development programs (*Alves, 2002*).

In tropical forests, a positive relationship is observed between the increase in both human population and natural resource extraction, and deforestation (*Laurance et al., 2002*; *Lopez-Carr & Burgdorfer, 2013*; *Lopez-Carr, Lopez & Bilsborrow, 2009*). However, in the Brazilian Amazon, this relationship is not always positive. While in some regions population density is not a direct cause of deforestation, in others it may be one of the leading causes (*Jusys, 2016*; *Tritsch & Le Tourneau, 2016*).

The age of the PA (or the time since its creation) is often correlated with better conservation results. Assessments in marine reserves reveal that areas that have been protected for longer show an increase in the quantity and richness of fish species (*Claudet et al., 2008*; *Molloy, McLean & Côté, 2009*). However, the relationship of PA age with conservation results may be antagonistic, with some younger PAs in the Brazilian Amazon obtaining better results in relation to reduction or avoidance of deforestation compared with older PAs (*Soares-Filho et al., 2010*).

The classification of PA classes according to the International Union for Conservation of Nature (IUCN) criteria (*Dudley, 2008*), into strictly protected (I–IV) and sustainable

use (or multiple use) management classes (V–VI), has generated several discussions on the efficiency of one category or another in reducing the illegal use of natural resources (*Nelson & Chomitz, 2011*). While some experts do not believe in the efficiency of multiple-use PAs in conserving biodiversity in the long term, others believe adoption of this class of PA will lead to a more effective and inclusive conservation strategy (*De Toledo et al., 2017*; *Schwartzman et al., 2010*).

*Laurance et al. (2012)* identified that in addition to the deforestation, across all three tropical continents, logging, wildfires, and overharvesting (hunting and harvest of non-timber forest products) are major threats to tropical PA integrity. Many of these threats, unlike deforestation, are difficult to detect (e.g., surface fire, small-scale gold mining, selective logging) or undetectable (e.g., hunting and exploitation of animal products and extraction of non-timber plant products) even with increasingly sophisticated remote sensing techniques (*Peres, Barlow & Laurance, 2006*). In this sense, on the ground enforcement activities can result in a wealth of information about the magnitude and types of illegal activities occurring within PAs (*Gavin, Solomon & Blank, 2010*) that are not detected by commonly used remote sensing techniques.

In this study, we evaluated the illegal use of natural resources within 118 federal PAs in the Brazilian Amazon, through the analysis of 4,243 illegal activities (infraction records) obtained from law enforcement activities in the period of 2010–2015. First, we categorized illegal activities to determine the main threats found within PAs. Then, we used the infraction records to evaluate the following hypotheses about the intensity of pressures on PAs from illegal activities: (a) fewer illegal activities occur in sustainable use PAs because they have fewer use restrictions than PAs under integral protection; (b) fewer illegal activities occur in older PAs because they have better established administrative structures and management than newer ones; (c) PAs with higher local population density tend to have more illegal activities because of greater anthropogenic pressure; and (d) PAs with greater accessibility tend to have more illegal activities.

## MATERIALS & METHODS

### Data sources

The data used as explanatory variables were obtained from the following publicly available sources: a shapefile describing the geographic boundaries of the Amazon biome from Ministério do Meio Ambiente (*MMA, 2016*); a shapefile describing the geographic boundaries of federal PAs (conservation units) from Instituto Chico Mendes de Conservação da Biodiversidade (*ICMBio, 2016b*); shapefiles describing water bodies (water masses) and rivers (multiscale ottocoded hydrographic base 2013) from Agência Nacional de Águas (*ANA, 2013*); a shapefile describing roads at 1:250,000, and limits of Brazil and South America from Instituto Brasileiro de Geografia e Estatística (IBGE) (*IBGE/DGC, 2015*); and shapefiles describing the populational "grid" of Brazil from IBGE (*IBGE, 2016a*).

The data on illegal use of natural resources (illegal activities) used were standardized and made available to authors by the Instituto Chico Mendes de Conservação da

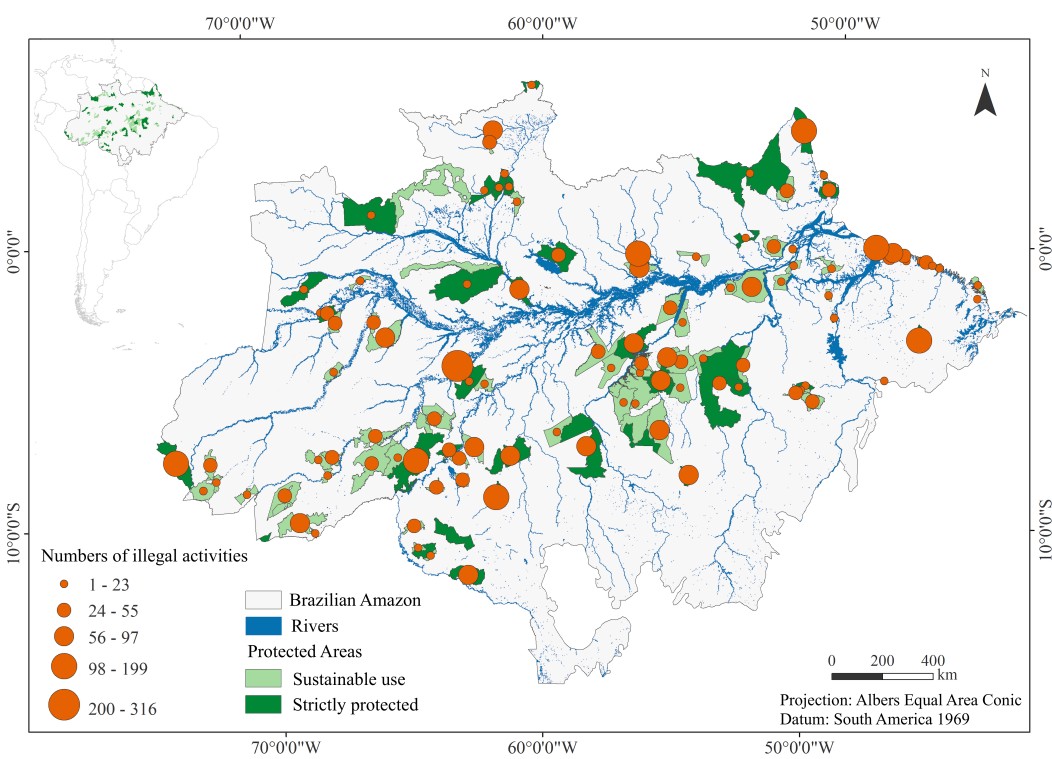

**Figure 1** **Illegal activities in the Brazilian Amazon federal protected areas.** Brazilian Amazon federal protected areas (sustainable use and strictly protected), and 4,243 occurrences (grouped per PA) of illegal use of natural resources (illegal activities) in the period of 2010–2015.

Biodiversidade/Divisão de Informação e Monitoramento Ambiental (ICMBio/DMIF, 2017, unpublished data; available upon request from ICMBio: http://www.icmbio.gov.br/portal/). The maps presented in this study (Fig. 1, Figs. S1 and S2) and area calculations were produced in an equal area projection (Projection: Albers Equal Area Conic; Datum: South America 1969). The geographic information system (GIS) environment was created and the elaboration of spatial variables performed based on geographic data obtained from official sources, in ArcGIS 10.2 software (ESRI, Redlands, CA, USA). The data on illegal activities compiled and formatted for our study are available in Data S1.

## Brazilian Amazon

We delimited the Brazilian Amazon (Fig. 1) according to the boundaries of the Amazonia biome as defined by the Instituto Brasileiro de Geografia e Estatística (*IBGE, 2004*). The IBGE's proposal follows the boundaries laid out in the original extension of the tropical rainforests of northern Brazil (*Góes Filho & Veloso, 1982*), which is inside the tropical moist broadleaf forests biome (*Olson et al., 2001*). The Brazilian Amazon covers an area of around 4.3 million km$^2$, about 50% of the of the country's territory. The region has a population of roughly 21.6 million people, 72% of whom live in cities in nine Brazilian states (Amazonas, Acre, Rondônia, Roraima, Amapá, Pará, Mato Grosso, Maranhão, and Tocantins) (*Silva, Prasad & Diniz-Filho, 2017*).

**Table 1  Summary of Brazilian Amazon federal protected areas included in the analysis.** Overall information about Brazilian Amazon federal PAs, IUCN category correspondence, absolute number of illegal activities and value of fines.

| Protected area class | Protected area category | IUCN | PAs ($n$) | Area (km$^2$) | Illegal activities ($n$) | Fines (U$)[a] |
|---|---|---|---|---|---|---|
| | Ecological Station | Ia | 10 | 55,248.94 | 257 | 27,594,947.29 |
| Strictly protected | Biological Reserve | Ia | 9 | 36,381.43 | 963 | 49,005,094.70 |
| | National Park | II | 19 | 204,324.04 | 959 | 67,348,814.39 |
| Subtotal | | | 38 | 295,954.42 | 2,179 | 143,948,856.38 |
| | Relevant Ecological Interest Area | IV | 3 | 189.31 | 6 | 13,573.23 |
| | Environmental Protection Area | V | 2 | 20,632.85 | 5 | 384,154.04 |
| Sustainable use | Sustainable Development Reserve | VI | 1 | 644.42 | 3 | 10,782.83 |
| | National Forest | VI | 32 | 164,262.20 | 901 | 54,675,627.21 |
| | Extrative Reserve | VI | 42 | 119,250.50 | 1,149 | 25,613,146.15 |
| Subtotal | | | 80 | 304,979.29 | 2,064 | 80,697,283.46 |
| Total | | | 118 | 600,933.71 | 4,243 | 224,646,139.84 |

Notes.

[a] All fines were imposed in Brazilian real (R$) and converted to American dollar (US$) by using an exchange rate of R$3.168: US $1 for the purpose of comparison with other studies. Dollar quotation on 03/31/2017.

## Federal protected areas

We evaluated 118 federal PAs established before 2010 in the Brazilian Amazon, totaling an area of around 600,000 km$^2$ (Fig. 1, Table 1, Table S1). Of these 118 PAs, 38 are strictly protected (Biological Reserve (Rebio), $n = 9$, IUCN Ia; Ecological Station (Esec), $n = 10$, IUCN Ia; and National Park (Parna), $n = 19$, IUCN II), and 80 are sustainable use (Area of Relevant Ecological Interest (Arie), $n = 3$, IUCN IV; Environmental Protection Area (Apa), $n = 2$, IUCN V; National Forest (Flona), $n = 32$, IUCN VI; Sustainable Development Reserve (RDS), $n = 1$, IUCN VI; and Extractive Reserve (Resex), $n = 42$, IUCN VI). Although fewer strictly protected than sustainable use reserves were analyzed, these two major classes of PA have similar total areas (strictly protected: roughly 295,000 km$^2$ and sustainable use: roughly 305,000 km$^2$). In total, we studied 91.5% of the PAs managed by the federal government in Amazonia, which corresponds roughly 76% of the total territory in federal PAs. Overall, Brazil have 789,280 km$^2$ distributed in 326 PAs managed by the federal government across the country and 127 PAs in Amazonia (ICMBio, 2016a).

All PAs are forested, with a few also featuring grasslands and savannas. Thirteen PAs are coastal/marine reserves. We excluded nine areas established after 2010, because we analyzed the period of illegal activity spanning from 2010 to 2015. In our study, only PAs (conservation unities) managed by the federal government under the Brazilian System of Conservation Units (SNUC) (Brasil, 2000) were evaluated. Therefore, for the purpose of this study, we excluded state, municipal and private areas, as well as indigenous lands and quilombola lands (traditional Afro-Brazilian communal territories).

## Illegal use of natural resources

Official figures for illegal use of natural resources (hereafter illegal activities) within federal PAs in the Brazilian Amazon were obtained by analysis of 4,243 environmental infraction records (Data S1, Table S1). Irregularities are framed according to Federal Decree 6514 (2008), which deals with administrative environmental infractions and

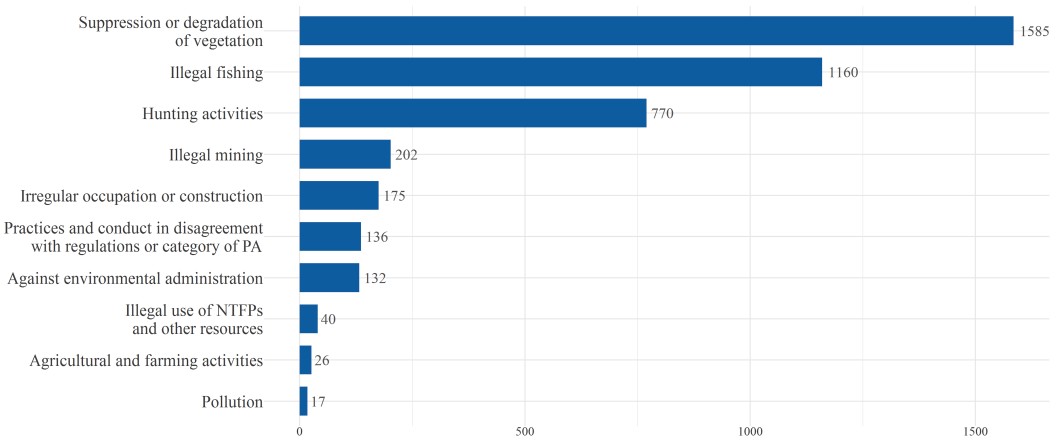

**Figure 2** **Illegal activities category and total number of occurrence.** Official figures for illegal use of natural resources (illegal activities) within federal PAs in the Brazilian Amazon obtained by analysis of 4,243 environmental infraction records. Categorization of illegal activities considered the infraction framework, the number of occurrences of each type of infraction (according to the Brazilian Federal Decree 6514 (2008)), and the main characteristics of illegal activities.

penalties (*Brasil, 2008*). For analytical purposes, we considered that each environmental infraction corresponded to an illegal activity.

Due to the large number of types of infraction and considering that the categories presented by the Brazilian Decree are very broad (e.g., hunting and fishing would fall into the same category), a new categorization of illegal activities was proposed. We considered the infraction framework, the number of occurrences of each type of infraction, and the main characteristics of illegal activities (Fig. 2, Table S2).

## Protected area accessibility

We defined accessibility (or accessible area) of PAs as the intersection between the total area of a PA with the area of a 10 km buffer adjacent to roads and rivers located within or outside PAs. The definition of accessibility within 10 km of rivers and roads takes into consideration that most natural resource exploitation in the Amazon is limited by transportation. Preliminary work conducted in Amazonia suggested that 10 km is the maximum distance local people can travel in order to collect non-timber forest resources and/or hunt (*Peres & Lake, 2003*; *Peres & Terborgh, 1995*).

To measure accessibility (Fig. S1, Table S2), we used the following procedures: creation of 10 km buffers around roads and rivers; union of the files produced when applying 10 km buffers; intersection of buffers and PAs (accessibility or accessible area); calculation of the accessible area (km$^2$); and division of the accessible area by the total area of the PA. All roads mapped by the IBGE at 1:250000 were considered (*IBGE/DGC, 2015*). Selection of the main rivers was carried out according to the criteria adopted by the National Water Agency for the characterization of Brazilian rivers, in which main rivers are drainage sections with an area of contribution greater than 20,000 km$^2$.

## Population density

Population density was considered at a distance of 50 km around the PAs. Population density information was obtained from the ''Brazilian statistical grid'' (*IBGE, 2016a*; *IBGE, 2016b*) prepared by IBGE based on the Brazilian population census of 2010 (*IBGE, 2010*; *IBGE, 2011*). The ''Brazilian statistical grid'' contains the amount of the Brazilian population in georeferenced polygons from 1 km$^2$ in rural areas and polygons up to 200 m$^2$ in urban areas. The grid is more refined than the municipal level data, which is generally used in studies that analyze demographic and socioeconomic factors for the Brazilian Amazon. For visualization purposes, we elaborated a population density map of the Amazon biome from the ''Brazilian statistical grid'' (Fig. S2).

In order to produce the population density variable (Table S2) in the area surrounding the PAs, we first created a 50 km buffer from the perimeter of each PA; then intersected the 50 km buffer area of each PA with the ''Brazilian statistical grid''; and finally divided the population within the buffer area of 50 km by its area (km$^2$). Areas located outside the Brazilian territory and in marine areas were excluded. When PAs were located very close to the border of the Amazon biome, a 50 km band was considered beyond the limits of the biome, but within Brazilian territory.

## Data analysis

A summary of all environmental infractions in the period from 2010 to 2015 allowed assessment of the main illegal uses of natural resources (by verifying the illegal activities that generated the infraction notices), as well as the categorization of these illegal uses (Fig. 2). The temporal trend of the illegal use of natural resources for the study period was evaluated using a linear regression. The total number of illegal activities was also summarized for each PA (Table S1), in relation to management categories (strictly protected and sustainable use) (Table 1). For further analysis, the three categories of illegal activities with the highest number of records and their totals summarized for each PA were used. In order to take in to account differences in the area of PAs and to standardize our variables, the total number of infractions and the total number of the three most common infraction categories were divided by the number of years ($n = 6$) and the area of the PA (km$^2$). This procedure was performed considering that the PAs have varied sizes and the measure of law enforcement effort that we adopted was the number of infraction records per year.

In order to normalize the data, transformations were applied to the following variables: illegal activities $= \log_{10} ((\text{illegal activities} \times 10^5) + 1)$; age $= \log_{10}$ protected area age; accessibility $= \sqrt{(\text{accessibility})}$; and population density $= \log_{10} (\text{population density} \times 10^5)$.

We used Spearman correlation analysis to evaluate the independence between our environmental variables (Table S3). Variables with weak correlations ($r_s < 0.50$) were retained for use in subsequent analyses. The differences in the influence of management classes of PAs (sustainable use or strictly protected), age, accessibility, and population density, on illegal activities occurring in PAs, were analyzed using generalized additive models (GAMs, Gaussian distribution family) (*Guisan, Edwards & Hastie, 2002*; *Heegaard, 2002*; *Wood, 2017*). GAMs were run separately for each of the three most recorded illegal activities. In order to verify possible differences in the number of illegal activities in stryctly

terrestrial PAs ($n = 105$) and coastal/marines ($n = 13$) ones, we used a Mann–Whitney $U$ test. All analyses were performed in the R environment for statistical computing (*R Development Core Team, 2016*).

## RESULTS

### Federal protected areas and illegal use of natural resources

Of the 118 PAs evaluated, 107 had at least one infraction reported between 2010 and 2015; only 11 had no records of illegal activities (Fig. 1, Table S1). Overall, there was a decrease in the number of illegal activities within federal protected areas in the Brazilian Amazon for the study period ($R^2 = 0.56$, $p = 0.09$). A total of 4,243 occurrences of illegal use of natural resources were evaluated, and these resulted in total fines of US\$224,646,139.84 (Table 1). Strictly protected PAs had a relatively higher total fines value (US\$143,948,856.38) compared to that of sustainable use reserves (US\$80,697,283.46). Similarly, strictly protected PAs presented slightly higher numbers of illegal activities ($n = 2,179$) than sustainable use reserves ($n = 2,064$). The mean number of total illegal activities in each PA was 35 (median 19.50), with 50% of PAs within the range of 8.0 to 47.5. The ten PAs with the highest frequency of illegal activities were Rebio do Abufari ($n = 316$), Parna Serra do Divisor ($n = 199$), Parna Mapinguari ($n = 187$), Rebio do Jaru ($n = 158$), Rebio do Gurupi ($n = 137$), Resex Marinha de Soure ($n = 129$), Parna do Cabo Orange ($n = 122$), Rebio Trombetas ($n = 122$), Flona do Jamaxim ($n = 97$), and Resex Chico Mendes ($n = 93$).

We found 27 types of illegal uses of natural resources that were grouped into 10 categories of illegal activities (Fig. 2, Table S2). The most commonly registered infractions were related to suppression and degradation of vegetation (37.36%), followed by illegal fishing (27.34%) and hunting activities (18.15%). These three categories together corresponded to 82.85% of all records of illegal activities in the entire study period. Infractions related to the suppression and degradation of vegetation were responsible for the highest total amount of fines among the 10 categories of illegal activities, US\$188,337,814.39, which corresponds to around 83% of all fines imposed in the study period. The four PAs with the highest number of illegal activities related to the suppression and degradation of vegetation were the Parna Serra do Divisor ($n = 109$), Rebio do Gurupi ($n = 94$), Parna Mapinguari ($n = 92$), and Resex Chico Mendes ($n = 71$). For illegal fishing, the Rebio do Abufari ($n = 168$), the Parna do Cabo Orange ($n = 120$), the Rebio Jaru ($n = 89$), and the Esec Maracá ($n = 52$), had the highest number of infractions. Regarding hunting, the four reserves with the majority of records were the Rebio do Abufari ($n = 168$), the Parna Serra do Divisor ($n = 72$), the Rebio Trombetas ($n = 46$), and the Flona Tefé ($n = 35$).

### Predictors of illegal activities within federal protected areas

The mean age of federal PAs in the Brazilian Amazon (calculated from 2015) was 18.92 years (median = 14, range = 6–54 years), with 50% of reserves ranging in age from 10 to 26 years. The total mean area of the PAs was 5,092.66 km$^2$ (median = 2,858.73 km$^2$). The reserves ranging from 1,209.90 to 6,813.01 km$^2$ in a 50 km buffer population density around PAs averaged 7.49 inhabitants per km$^2$ (median = 1.54 inhabitants/km$^2$), with 50% of the PAs ranging from 0.63 to 4.68 inhabitants per km$^2$. The protected area with the lowest

**Table 2 Generalized additive models (GAMs) results.** Parameter (Slope) estimates of explanatory variables from the GAMs on the number of illegal activities in the Brazilian Amazon federal PAs.

| | All illegal activities[a] | | Hunting activities[b] | | Illegal fishing[c] | | Vegetation degradation[d] | |
|---|---|---|---|---|---|---|---|---|
| | Slope (SE)[e] | t value | Slope (SE)[e] | t value | Slope (SE)[e] | t value | Slope (SE)[e] | t value |
| Intercept | −1.460 (0.521) | −2.80[***] | −0.909 (0.577) | −1.57 | −1.718 (0.643) | −2.67[***] | −1.318(0.618) | −2.13[**] |
| Classes[f] (Sustainable use) | −0.160 (0.159) | −1.00 | −0.198 (0.176) | −1.12 | −0.476 (0.197) | −2.42[**] | 0.163 (0.189) | 0.86 |
| Age[g] | −0.005 (0.007) | −0.67 | 0.002 (0.008) | 0.29 | −0.005 (0.009) | −0.53 | −0.009 (0.009) | −1.02 |
| Accessibility[h] | 0.968 (0.256) | 3.85[****] | 0.502 (0.284) | 1.77[*] | 0.899 (0.316) | 2.84[***] | 0.527 (0.304) | 1.73[*] |
| Population density[i] | 0.574 (0.112) | 5.14[****] | 0.317 (0.124) | 2.56[**] | 0.508 (0.138) | 3.68[****] | 0.449 (0.133) | 3.39[****] |
| R-square adjusted[j] | 0.391 | | 0.116 | | 0.284 | | 0.194 | |
| Model deviance explained (%)[k] | 41.20 | | 14.60 | | 27.30 | | 22.10 | |
| Model GCV[l] | 0.541 | | 0.665 | | 0.826 | | 0.763 | |

**Notes.**

[*]$p < 0.10$.

[**]$p < 0.05$.

[***]$p < 0.01$.

[****]$p < 0.001$.

[a]Includes the total of all illegal activities (illegal activities/protected area size km$^2$/number of years) log transformed (($\log_{10} + 1$) $\times 10^5$).

[b]Includes all hunting activities (hunting activities infractions/protected area size km$^2$/number of years) log transformed (($\log_{10} + 1$) $\times 10^5$).

[c]Includes all illegal fishing (illegal fishing/protected area size km$^2$/number of years) log transformed (($\log_{10} + 1$) $\times 10^5$).

[d]Includes all vegetation degradation (vegetation degradation/protected area size km$^2$/number of years) log transformed (($\log_{10} + 1$) $\times 10^5$).

[e]Slopes for variables and Standard Error (SE).

[f]Class of protected areas (Sustainable use and Strictly protected).

[g]Age of protected area creation (creation until 2015) log transformed ($\log_{10}$).

[h]Accessibility of protected area square root transformed.

[i]Population density in a 50 km buffer from the perimeter of each PA log transformed ($\log_{10} \times 10^5$).

[j]R-square adjusted for each model.

[k]Percentage of Deviance Explained for each model (%).

[l]Generalized Cross-Validation score for each model (GCV).

population density in the surroundings was the Resex do Xingu with 0.06 inhabitants/km$^2$ and the highest density was found in the neighborhood of Parna Anavilhanas with 75.90 inhabitants/km$^2$. The overall index of accessibility was on average 43% (median $= 33\%$), and 50% of PAs had accessibility between 15% and 68%. Regarding accessibility, it is important to highlight that 17 PAs presented 100% of this variable, as well as 10 PAs had zero accessibility (Table S1).

The explanatory power of the GAMs was low for all groups (Table 2), with a maximum explained variation of 41.20% ($R^2$ adjusted $= 0.39$) for total illegal activities, and a minimum of 14.6% ($R^2$ adjusted 0.12) for illegal hunting activities. From all explanatory variables analyzed in our study, population density was the most important predictor of total number of infractions (Fig. 3), as well as illegal fishing, suppression and degradation of vegetation, and hunting. The second most important predictor of illegal activities was accessibility, which was positively related to all illegal activities (Fig. 4) and to illegal fishing. PA classification was only an important predictor for illegal fishing, with sustainable use PAs having lower levels of illegal fishing. The age of a PA was not a significant predictor for any of the illegal activities analyzed in our study.

In relation to the number of illegal activities and the PA location (coastal/marine or terrestrial), we found a significant decrease in the number of all illegal activities ($p < 0.001$)

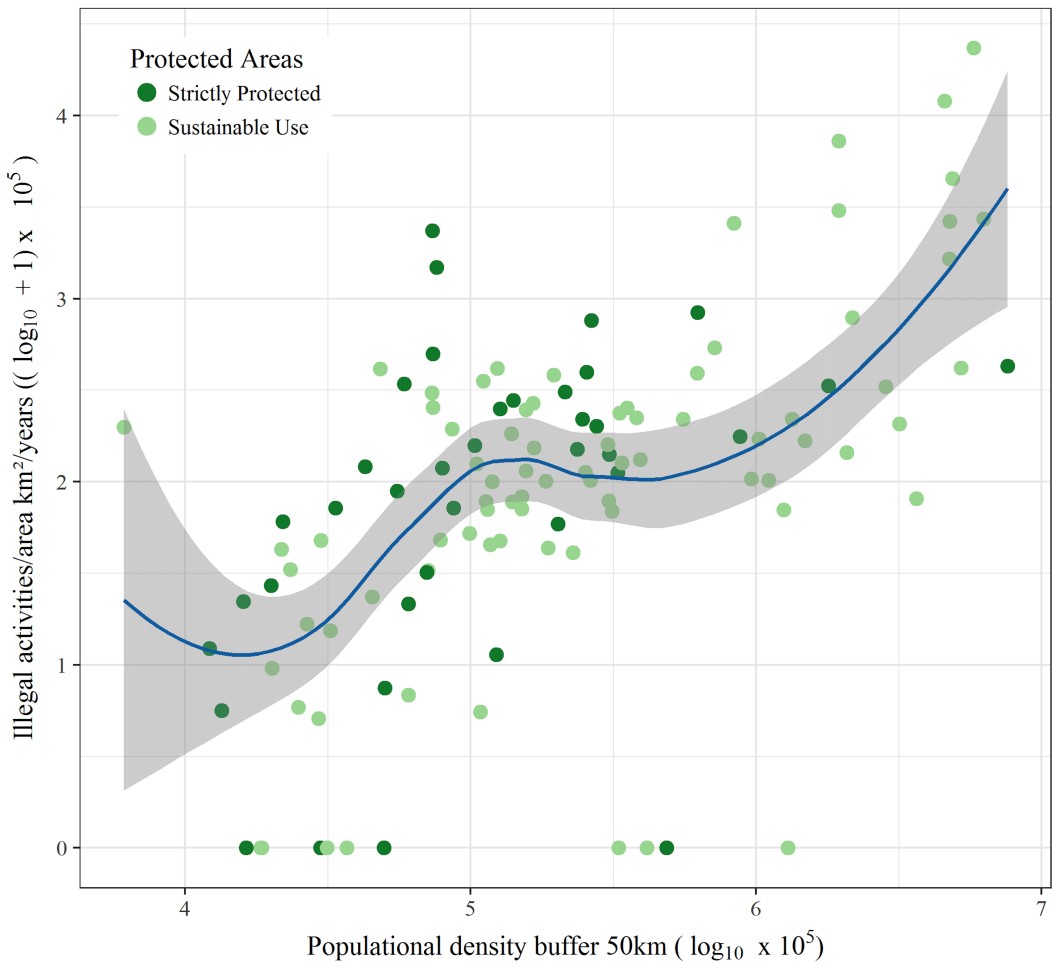

**Figure 3** Total of all illegal activities and human population density in a 50 km buffer from the perimeter of each protected area.

and a significant increase in the number of illegal fishing ($p < 0.001$) in coastal/marine PAs (Table S4). Illegal activities related with hunting and vegetation degradation were not significantly different in these two locations of PAs.

## DISCUSSION

Globally, the illegal use of natural resources is one of the biggest threats to biodiversity, and generally threatens the integrity of PAs and the viability of endangered species (*Conteh, Gavin & Solomon, 2015*; *Dinerstein et al., 2007*; *Gavin, Solomon & Blank, 2010*; *Laurance et al., 2012*). Despite the fact that Amazonian PAs are one of the most important means of reducing deforestation rates in the biome (*Kere et al., 2017*), PA creation alone is not sufficient to reduce threats to biological diversity.

Our analysis showed that there was a wide range of illegal activities that threatens the biodiversity of Amazonian federal PAs. We found that illegal activities related to suppression and degradation of vegetation, illegal fishing and hunting activities were the most

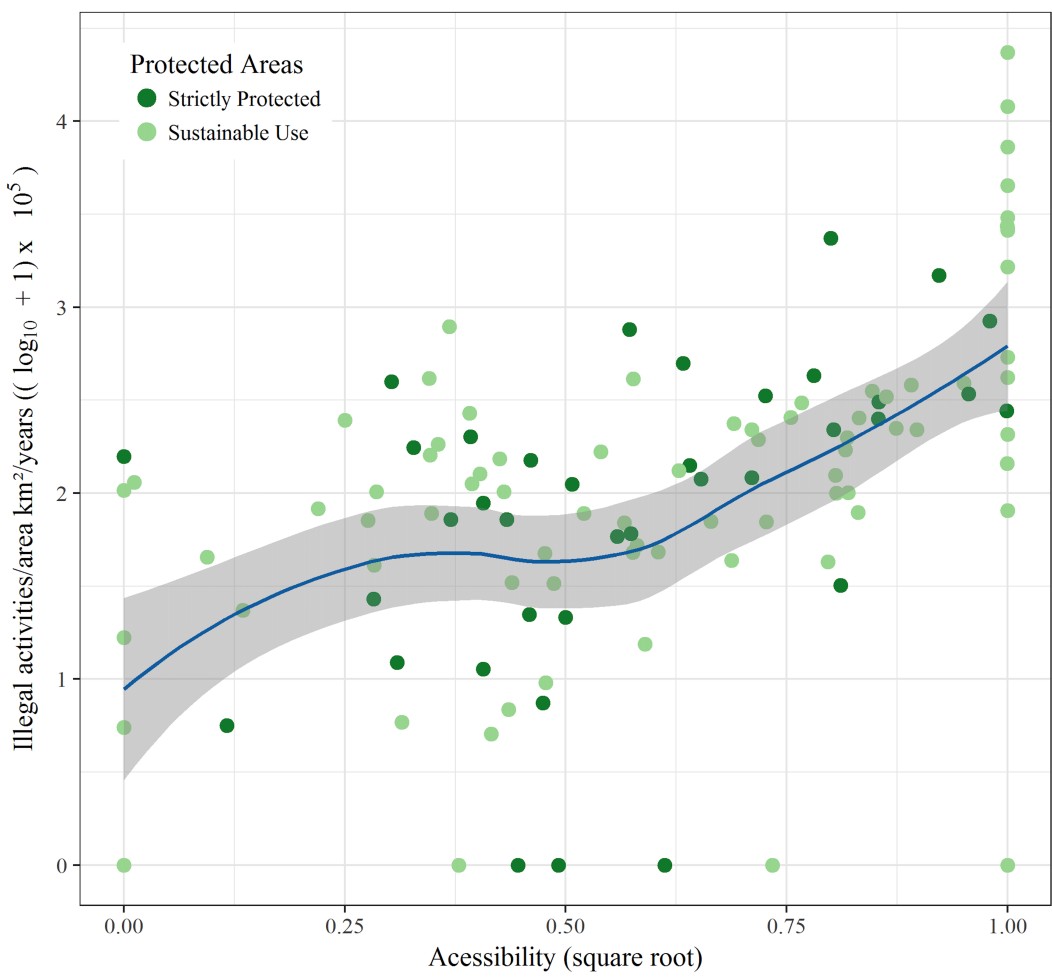

**Figure 4** Total of all illegal activities and accessibility of protected areas.

commonly recorded. These three activities have been highlighted in several assessments of biodiversity threats globally: hunting and the illegal wildlife trade (*Dudley, Stolton & Elliott, 2013*; *Nijman, 2015*; *Sharma et al., 2014*; *Tella & Hiraldo, 2014*; *Underwood, Burn & Milliken, 2013*); fishing in prohibited locations, outside permitted periods and in excess of established quantities or sizes (*Free, Jensen & Mendsaikhan, 2015*; *Sethi & Hilborn, 2008*; *Thomas, Gavin & Milfont, 2015*); and illegal logging, deforestation and degradation of vegetation (*Chicas et al., 2017*; *Curran et al., 2004*; *Funi & Paese, 2012*; *Yonariza & Webb, 2007*). Although illegal activities related to the suppression and degradation of vegetation, illegal fishing, and poaching activities were those most frequently recorded in Amazonian PAs, it does not mean that other less prominent illegal activities are not of concern.

The population density surrounding PAs was the most important variable in our study, predicting total illegal activities, as well as the suppression and degradation of vegetation, illegal fishing, and poaching activities. This finding is in line with the results of other tropical forest studies that have observed a positive relationship between the growth of human populations and an increase in natural resource extraction and deforestation

(*Geist & Lambin, 2002*; *Laurance, Sayer & Cassman, 2014*; *Lewis, Edwards & Galbraith, 2015*; *Lopez-Carr & Burgdorfer, 2013*; *Marques, Schneider & Peres, 2016*).

We found that accessibility was positively related only with the total number of illegal activities and to illegal fishing, while for hunting activities and vegetation suppression and degradation activities this variable was marginally significant. Despite this, it was possible to verify the importance of accessibility in predicting illegal activities within PAs. Roads and highways have a fundamental role in opening the tropics to destructive colonization and exploitation (*Laurance et al., 2001*). Roads provide access and dispersion of people within tropical forests and facilitate access for hunters, miners, land speculators, and others into forest core areas (*Laurance, Goosem & Laurance, 2009*). For example, the increasing deforestation of the Brazilian Amazon began with the construction of the Belém-Brasília highway in the 1960s (*Vieira et al., 2008*) and the opening of the Transamazon Highway in 1970 (*Fearnside, 2005*). *Barber et al. (2014)* observed that until 2006, deforestation in the Brazilian Amazon was higher in areas closer to roads and rivers, with almost 95% of the total deforested area within 5.5 km of roads and up to 1 km from rivers. Recent studies show that populations of aquatic species (e.g., giant otters, alligators) in more accessible areas have collapsed throughout the Amazon basin (*Antunes et al., 2016*).

We found no relationship between the age of PAs and illegal activities, although the age of a PA is often correlated with conservation results (*Claudet et al., 2008*; *Molloy, McLean & Côté, 2009*; *Soares-Filho et al., 2010*). Our results show that sustainable use PAs decrease the frequency of illegal fishing activities. This relationship can be attributed to the fact that residents of the reserves assist surveillance. *Nepstad et al. (2006)* verified that sustainable use PAs and indigenous lands hold great importance for the prevention of deforestation and wildfires. This pattern was also observed in a global analysis of the effectiveness of strictly protected and sustainable use PAs in reducing tropical forest fires, where sustainable use PAs were more efficient (*Nelson & Chomitz, 2011*). *Porter-Bolland et al. (2012)* observed that forests managed by communities presented lower and less variable deforestation rates across the tropics. These findings reinforce the idea that in order to achieve an effective conservation, it is necessary to involve local communities in environmental governance (*Brondizio & Le Tourneau, 2016*; *Dudley et al., 2014*).

Despite differences found in the decrease in the number of total illegal activities and the increase in illegal fishing activities in coastal/marine when compared with terrestrial PAs, we did not find significant differences for illegal activities of hunting and vegetation degradation. Overall, a greater number of fisheries-related offenses are expected in coastal marine areas. However, coastal marine PAs that occur in the Brazilian Amazon have also significant portions of forests (mainly mangrove formations). Thus, it is not surprising that illegal hunting and vegetation degradation were present in these areas in similar levels of terrestrial PAs. On the other hand, the differences presented here indicate the need for a more detailed evaluation of these different locations of PAs, which could be coupled with differences in strategies and conservation actions to be applied to individual areas (*Barber et al., 2012*; *Margules & Pressey, 2000*).

## CONCLUSIONS

PAs are fundamental for biodiversity conservation across the Brazilian Amazon, and their establishment and maintenance is a key strategy for protection from the pressures and threats posed by human presence in tropical forests. Nonetheless, PAs are one of the most crucial factors contributing to reductions in deforestation in this biome. We report several threats that may impair long-term conservation and many efforts are still needed to address these issues. The use of enforcement reports generated by official government authorities provides us with a more nuanced view of the illegal activities taking place within PAs in the Brazilian Amazon. We demonstrated that this type of information can be useful as a complement to more sophisticated remote sensing techniques that usually fail to identify threats under the forest canopy. We have showed that the monitoring information helps to identify more problematic PAs in relation to the illegal use of natural resources and in relation to detailed categories of infraction. This can help managers to plan and implement specific conservation actions to individual areas in order to reduce illegal activities. Additionally, information regarding enforcement effort applied in each PA can be better quantified, which would help conservationists and practioners to be able to evaluate and set goals for different PAs under different regimes and locations. Implement management actions in and around PAs are key conservation issues that will need to be addressed to provide the realization of effectiveness goals of *de facto* preservation of the Brazilian Amazon.

## ACKNOWLEDGEMENTS

We wish to thank DMIF/CGPRO/ICMBio for providing access to illegal activities (fines) recorded within the Federal Protected Areas, in special for the ICMBio environmental analysts Kelly Borges, Mariella Butti, and Andre Alamino. We would like to thank Luis Barbosa for assistance with some GIS procedures.

### Funding

Fernanda Michalski received a productivity scholarship from CNPq (Process 301562/2015-6). José Maria Cardoso da Silva was supported by the University of Miami and Swift Action Fund. Érico Emed Kauano was supported by Instituto Chico Mendes de Conservação da Biodiversidade. This work was funded by a research grant from the CAPES Foundation, Ministry of Education of Brazil (Project 88881.030414/2013-01). There was no additional external funding received for this study. The funders had no role in study design, data collection and analysis, decision to publish, or preparation of the manuscript.

### Grant Disclosures

The following grant information was disclosed by the authors:
CNPq: 301562/2015-6.
University of Miami and Swift Action Fund.

Instituto Chico Mendes de Conservação da Biodiversidade.
CAPES Foundation.
Ministry of Education of Brazil.

**Competing Interests**

The authors declare there are no competing interests.

**Author Contributions**

- Érico E. Kauano conceived and designed the experiments, performed the experiments, analyzed the data, contributed reagents/materials/analysis tools, wrote the paper, prepared figures and/or tables, reviewed drafts of the paper.
- Jose M.C. Silva conceived and designed the experiments, wrote the paper, reviewed drafts of the paper.
- Fernanda Michalski conceived and designed the experiments, analyzed the data, wrote the paper, prepared figures and/or tables, reviewed drafts of the paper.

**Data Availability**

The raw data has been provided as a Supplemental File.

**Supplemental Information**

Supplemental information for this article can be found online at http://dx.doi.org/10.7717/peerj.3902#supplemental-information.

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
