# Peer review of "Illegal use of natural resources in federal protected areas of the Brazilian Amazon"

_PeerJ, doi:10.7717/peerj.3902_

## Round 0.1 · original submission · Minor Revisions

Please follow the reviewers suggestions carefully, specially improving figures legends and the conclusion section. Do not repeat your results, Try to use this section to enhance them, as one of the reviewers suggest explain for example how your findings will improve government measurments to prevent biodiversity losses.

Reviewer 1 ·

Basic reporting

no comment

Experimental design

no comment

Validity of the findings

I find the overall findings valid, however I believe it would be interesting to add in the analysis the temporal trend of the illegal use of natural resources and if this changes over time i.e. if there is an increase or decrease. Moreover, I wonder if the authors have looked at the correlation between the illegal use of natural resources and law enforcement activity (number of guards, monthly patrol etc) . Beside the PA protection category, does the presence of law enforcement activity have an impact on the number of illegal activities over time? PA protection category is not always correlated with law enforcement effectiveness.

These additional analyses would help to understand why there so many illegal activities and if law enforcement and/or the park management is efficient and effective. The take-home message would be beneficial for both managers and decision-makers. Moreover, I have an analytical concern about including both terrestrial and coastal/marine reserves – I expect that fishing activity is more present in coastal/marine areas and hunting activity more present in terrestrial areas. It would be good if this could be explored and included in the analysis.

There are many repetitions in the discussion section about the description of the results (the result section already includes it). This needs to be better structured. Moreover, I would suggest to add more literature about protection measures against illegal use of natural resources.

Additional comments

This manuscript was nicely written and interesting to biodiversity conservation and protected areas management. I have highlighted a few areas that could benefit from some clarification or rewording, which are listed below:

L123. IBGE, could you give the whole word Instituto Brasiliero de Geografia e Estatistica
L131. Established is repeated twice
L140-141. Can the authors add the total area (in square km) of the PAs studies overall the ones that are managed by the federal government?
L142. Can the authors add somewhere the percentage of grassland and savanna per park? Would be good a table with the PAs and the % of habitat cover types
L145 Brazilian System of Conservation Units
L146-148. Information not necessary
L181. 1 Km2 instead of from 1km by 1 km
L181. 200 m2 instead of up to 200m by 200m
L207. Please add table with results from correlations
L215. Would be nice have subtitles like in the methodology section
L271. Threatens
L294. Reference missing
L314…in order to achieve just an effective conservation, it is necessary to involve local…

Table S2. Please add a clear description of each category of illegal activity

Reviewer 2 ·

Basic reporting

The manuscript’s content is relevant given the lack of studies that contemplate the illegal use of natural resources in the PAs. This kind of research is important to understand the PA's needs and to enable some strategic actions.

The MS was very well written, and brought a lot of good and recent references.

Experimental design

Some of the reviewer suggestions to improve the MS are bellow :

Line 43: Change "significant reduction in deforestation in the region" to "significant deforestation reduction in the region"

Line 117: Put the Datum used to compose the maps.

Line 126: Change "of the territory of the country" to "of the country's territory"

Line 131: Take away one the words "established"

To all figures: Put the Datum and Projection in the maps, with this anyone that reads your maps will know this information.

Figure 1:
In the legend: put the units of the “Illegal activities”, like: “Numbers of illegal activities”
Divide the legend in two columns, the map’s layout will be improved.
Suggestion to the legend order an items:

First column:
Numbers of Illegal activities
1-23
24-55
56-97
98-199
200-316

Second column:
Brazilian Amazon
Rivers
Protected Areas
Sustainable use
Strictly protected


To all tables:Change “Brazilian reais” to “Brazilian real”

Validity of the findings

The results show the importance of your work, however i missed some explanation in the conclusions, that how the government authorities would use this data to improve the
threats to the biodiversity in the PAs.

---

## Round 0.2 · accepted · Accept

The authors have included the suggestions made by the reviewers and by the editor